# Common communicable diseases in the general population in France during the COVID-19 pandemic

Titouan Launay[1]*, Cécile Souty[1], Ana-Maria Vilcu[1], Clément Turbelin[1], Thierry Blanchon[1], Caroline Guerrisi[1], Thomas Hanslik[1], Vittoria Colizza[1], Isabelle Bardoulat[2], Magali Lemaître[2], Pierre-Yves Boëlle[1,3]

1 Sorbonne Université, INSERM, Institut Pierre Louis d'Épidémiologie et de Santé Publique, Paris, France, 2 IQVIA, Paris, France, 3 Hôpital Saint-Antoine, Assistance Publique–Hôpitaux de Paris, Paris, France

* titouan.launay@iplesp.upmc.fr

**Data Availability Statement:** The acute gastroenteritis and the chickenpox incidence are available from the Sentinelles website: https://www.sentiweb.fr/france/fr/?page=table.

## Abstract

In France, social distancing measures have been adopted to contain the spread of COVID-19, culminating in national Lockdowns. The use of hand washing, hydro-alcoholic rubs and mask-wearing also increased over time. As these measures are likely to impact the transmission of many communicable diseases, we studied the changes in common infectious diseases incidence in France during the first year of COVID-19 circulation. We examined the weekly incidence of acute gastroenteritis, chickenpox, acute respiratory infections and bronchiolitis reported in general practitioner networks since January 2016. We obtained search engine query volume for French terms related to these diseases and sales data for relevant drugs over the same period. A periodic regression model was fit to disease incidence, drug sales and search query volume before the COVID-19 period and extrapolated afterwards. We compared the expected values with observations made in 2020. During the first lockdown period, incidence dropped by 67% for gastroenteritis, by 79% for bronchiolitis, by 49% for acute respiratory infection and 90% for chickenpox compared to the past years. Reductions with respect to the expected incidence reflected the strength of implemented measures. Incidence in children was impacted the most. Reduction in primary care consultations dropped during a short period at the beginning of the first lockdown period but remained more than 95% of the expected value afterwards. In primary care, the large decrease in reported gastroenteritis, chickenpox or bronchiolitis observed during the period where many barrier measures were implemented imply that the circulation of common viruses was reduced and informs on the overall effect of these measures.

## Introduction

The first case of coronavirus disease 2019 (COVID-19) was reported in France January 24th, 2020 [1]. Investigation of local transmission clusters started from this date on yet no measures were directed to reduce transmission in the general population. In early March 2020, the observation of an increased circulation of the disease by surveillance systems and in hospital

**Funding:** IQVIA provided support in the form of salaries for authors ML and IB, but did not have any additional role in the study design, data collection and analysis, decision to publish, or preparation of the manuscript. The specific roles of these authors are articulated in the 'author contributions' section.

**Competing interests:** The authors have read the journal's policy and have the following competing interests: ML and IB are employees of IQVIA France. This does not alter our adherence to PLOS ONE policies on sharing data and materials. There are no patents, products in development or marketed products associated with this research to declare.

admissions [2–4] led to stronger interventions and the promotion of standard barrier measures. A first national lockdown was adopted from March 17th to May 11th, during which mobility was restricted, schools, public services and shops were closed and telework widely adopted. At the end of the lockdown a national strategy of testing/tracing/isolating was implemented. The circulation of the COVID-19 remained at low levels during the summer months but showed renewed activity from the beginning of September and eventually led to a second lockdown from October 18th, to December 15th 2020.

Transmission of SARS-CoV-2 is primarily via the respiratory route through direct or indirect contacts [5]. These transmission routes are the same for other respiratory viruses. Standard precautionary measures should therefore limit transmission for all these diseases. Indeed, mask wearing is thought to have an impact on the circulation of these viruses by limiting the emission of droplets [6, 7]. Handwashing may reduce direct transmission and contamination of surfaces, additionally limiting the transmission of viruses associated with the oro-fecal route. Other barrier measures, including physical distancing, may further reduce transmission [6, 7].

In France, the adoption of standard precautionary measures has been widely publicized with the population and highly adopted. Mask wearing was rare before March 2020, but the proportion of individual wearing a mask in public place increased from 25% to 58% between March 22 and April 20, 2020 [8]. Likewise the practice of handwashing increased, peaking during the first lockdown but decreased afterwards [8].

Here we aimed at describing how the spread of common communicable diseases changed during the COVID-19 epidemic in France. We focused on chickenpox, acute diarrhea, acute respiratory infections and bronchiolitis for which pre-COVID-19 surveillance was available. We used several data streams to investigate changes associated with different periods in the COVID-19 epidemic.

## Materials and methods

### Sources of data

All data were collected prospectively from January-2016 to December 2020 and presented by week (using the ISO-8601 system). When necessary, the computations described below were repeated by age groups or by administrative region.

During the pandemic, the general population was encouraged to consult with their GPs as usual [9]. No triage system for COVID-19 patients was put into place. Importantly, teleconsultations by phone or internet were reimbursed as normal visits.

Incidence of acute diarrhea (AD) and varicella was obtained from the Sentinelles network. The Sentinelles network has been monitoring common illnesses in primary care in France over the past 30 years [10]. Approximately 500 volunteer general practitioners (GPs, or about 1% of French GPs) take part in the surveillance system. Each week, GPs report the number of patients consulting for nine diseases or medical conditions. The case definition of AD was "at least three daily watery or nearly so stools, dating less than 14 days". The case definition for chickenpox was "typical eruption (erythemato-vesicular during 3 to 4 days with a dessication phase) beginning suddenly, with moderate fever (37˚5C – 38˚C)".

Incidence for Acute Respiratory Infection (ARI) and bronchiolitis was obtained from IQVIA's EMR database. This database collects medical information directly from electronic medical records from 1,200 GPs in France (approximately 2% of all French GPs). The panels of contributing GP are maintained as a representative sample of the primary care physician population according to 3 criteria known to influence prescribing: age, sex, and geographical distribution. Additionally, the patient population is representative of the country population

according to age and gender distribution, as provided by national statistic authorities. For each medical consultation, anonymized patient data are collected, including gender, year of birth and diagnosis (ICD-10 code). Incidence in the IQVIA and Sentinelles network are based on clinical diagnoses. Additionally, in the Sentinelles network, a random subset of patients presenting with ARI symptoms undergo nasopharyngeal swabs followed by RT-PCR to test for 4 viruses (influenza, human metapneumovirus, respiratory syncytial virus and rhinovirus), and for SARS-CoV-2 since March 2020 [11].

Sales of selected drugs were obtained from IQVIA Pharmaone LMPSO ("Le Marché Pharmaceutique Sell Out"). This database provides prescription and over-the-counter drugs dispensing data from 14,000 pharmacies in France (60% of all French pharmacies). The data collected includes Ephmra ATC classification and the number of boxes sold per week and per region. We selected drugs frequently prescribed in the case of AD by ATC code level 3/4, including A03A (plain antispasmodic and anticholinergics), A07B (intestinal absorbent antidiarrheals), A04A9 (others antiemetics and antinauseants) and A07H (mobility inhibitors) [12].

Search query volume in the google search engine were obtained from the Google Trends website [13]. We looked for the keyword "gastro" which is commonly used for AD in France, "bronchiolite" and "varicelle" which are French for bronchiolitis and chickenpox. The keywords "gastro" and "varicelle" are known to have a high correlation with the incidence of acute diarrhea and chickenpox respectively [14]. Searches were limited to the French language and from locations in France.

The GrippeNet surveillance system provided the percentage of patients consulting with a general practitioner in case of influenza-like illness (ILI) symptoms. GrippeNet is a population-sourced surveillance system where over 6000 participants report current symptoms every week and medical consultations or treatment associated with these symptoms [15]. The ILI definition used in GrippeNet.fr is the combination of three criteria: the sudden onset of symptoms, at least one of the following signs: fever, chills, headache, myalgia or asthenia and at least one of the following respiratory symptoms: cough, sore throat, dyspnea.

We obtained data on the number of individuals hospitalized with COVID-19 from the French national site for open data [16] where information was available by day and administrative region. Incidence of hospitalization was computed with respect to the population of the corresponding region.

## Statistical methods

Incidence of ARI, AD, chickenpox and bronchiolitis shows strong seasonal activity patterns [17, 18]. To describe changes from usual activity patterns, we split the data into two periods the "pre-COVID-19 period" (2016–2019) and the "COVID-19 period" (year 2020). A model was fitted to disease incidence, drug sales and search query volume in the pre-COVID-19 period using a linear trend and periodic regression terms as follows:

$$\log\left(y(w)\right) = a_0 + a_1 \times w + a_2 \times \cos\left(2\,\pi\frac{w}{52} + \varphi_1\right) + a_3 \times \cos\left(4\,\pi\frac{w}{52} + \varphi_2\right) + \varepsilon(w)$$

where $y(w)$ is the modelled quantity for week $w$ and $\varepsilon(w)$ a normally distributed random error. The coefficients of the regression were estimated by least squares minimization. Following the "Serfling" method used in the Sentinelles network [19], the fitted model was used to predict the expected value of $y_{exp}(w)$ over year 2020 as well as a 95% expected interval: $y_{exp}(w)\pm1{,}96\times s$ where $s$ is the residual standard deviation.

Year 2020 was split in several periods as indicated in Table 1, corresponding to changes in the national strategy to control the spread of SARS-CoV-2 [20] and summer vacations. We

**Table 1. Time periods defined according to the adoption of mitigation measures against COVID-19 and summer vacations.**

|  | Pre-lockdown | Lockdown | Post-lockdown | Summer Vacation | Autumn | Lockdown2 |
|---|---|---|---|---|---|---|
| Dates | 1/1–17/3 | 17/3-11/5 | 11/5-6/7 | 6/7-1/9 | 1/9-14/10 | 14/10-15/12 |
| Weeks | 1–12 | 13–20 | 21–28 | 29–36 | 37–42 | 43–51 |
| Limit mass gatherings |  | ■ | ■ | ■ |  | ■ |
| School/Daycare closure |  | ▨ | ▨ |  |  |  |
| Stay-at-Home / Teleworking / Public place closure |  | ■ | ▨ |  |  | ■ |
| Mask wearing |  | ▨ | ■ | ■ | ■ | ■ |
| School holiday |  |  |  | ■ |  |  |

Dark color corresponds with national implementation, light color with partial/regional implementation and white with no implementation.

computed the relative reduction between expected and observed values during each period as $RR = 1 - \frac{\Sigma_y(w)}{\Sigma_{y_{exp}}(w)}$ where the sum was over the weeks in each period.

## Results

### Acute diarrhea

The strong seasonal variations in all 3 indicators of AD (incidence, search query volume and drug sales) in the pre-COVID-19 years were well captured by the models ($R^2$ = 0,79). Observed values for all weeks in the 2016–2019 period remained in the prediction intervals.

In year 2020, the 3 indicators for AD showed a similar change over time. AD Incidence was high in January, corresponding to a typical winter-time increase. AD incidence was close to the expected level in February, well within the 95% prediction limits. However, incidence dropped rapidly in the second half of March, synchronously with the national lockdown adoption in France (March 17th–week 12). A similar pattern was found for web-search engine queries and drug sales, with a drop synchronous with the lockdown (Fig 1). The AD incidence changed from 94/100,000 inhabitants (95% CI, [84, 104]) in week 11 to 25/100000 (95% CI, [20, 30]) in week 15. In this latter week, the reduction was 75% with respect to the expected incidence (101 (95% CI, [76, 136])). The overall pattern of change for AD series is summarized in Table 2 for all periods of interest. During the lockdown (March 17th–May 11th), AD incidence was 67% lower than expected. The reduction was smaller in the subsequent period (37%) and during the summer vacation (34%), but increased again during the second lockdown. The largest reduction was for children (0–14 years old) and young adults (15–44) (S2 Fig).

Drug sales showed a decrease during the first lockdown and returned closer to the expected level at the end of the summer vacation, remaining in the lower part of the expected interval throughout. For search query volume, the data returned to normal during the post-lockdown and summer vacation period, and decreased again in the second lockdown period (Fig 1).

There was some regional variation in the relative reduction in AD incidence: during the first lockdown, the reduction ranged from 35% to 75% (S3 Fig). We found that regions where COVID-19 circulation had been the largest in the first wave experienced the largest reduction in AD incidence (Pearson's correlation coefficient with number of hospitalizations r = 0,59).

### Chickenpox

The data for chickenpox for the pre-COVID-19 period (2016–2019) showed seasonal variations that were well captured by the models ($R^2$ = 0,75). In 2020, chickenpox incidence rates in general practice were close to the expected level in the beginning of the year, and presented a

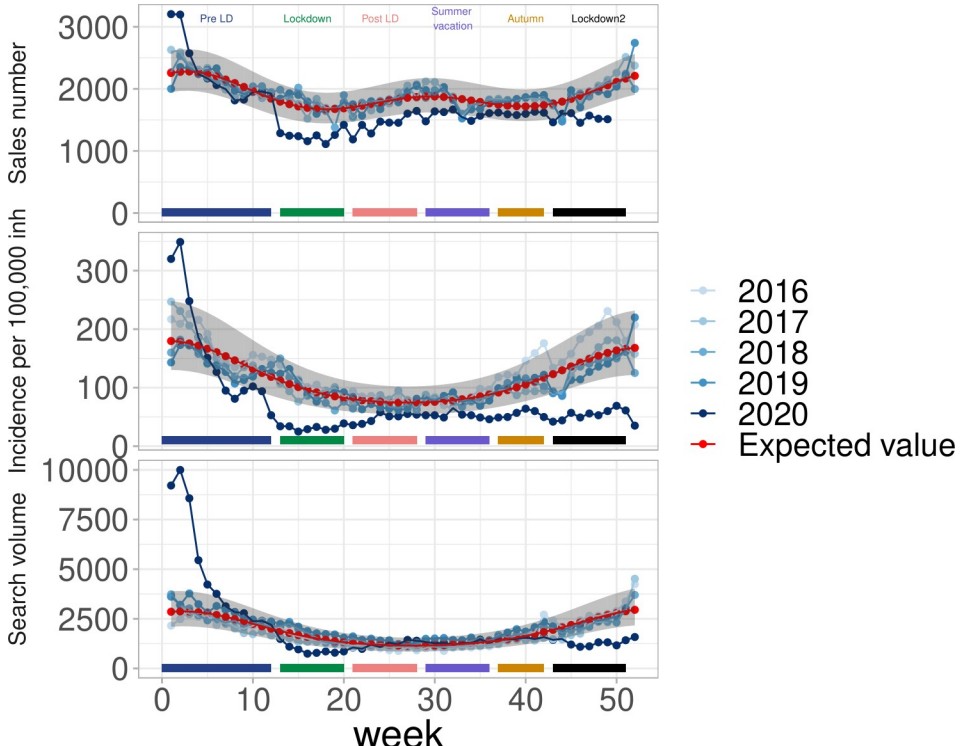

**Fig 1. Acute diarrhea in France during the COVID-19 pandemic.** (top) drug sales, (middle) Incidence in general practice and (bottom) search query volume.

sharp decrease during the first lockdown period (Fig 2). They were close to 0 during the whole first lockdown period and remained very low afterwards. Incidence rates were close to the expected level during the summer vacation, where disease incidence is always small and showed a renewed increase in the autumn period, although remaining close to the expected level. Overall, incidence rates dropped by 90% with respect to the expected level during the first lockdown period and by 97% during the post-lockdown (Table 2). The number of search engine queries showed similar changes over time (Fig 2), although the decrease compared to the expected level was less pronounced during the first lockdown period and a renewed reduction was observed in the two last periods (autumn and second lockdown).

The relative reduction in chickenpox incidence showed regional variation during the first lockdown, the reduction ranged from 50% to 90% (S4 Fig). No significant correlation was found between the chickenpox reduction and COVID-19 hospitalizations.

## Bronchiolitis

The seasonal variation of bronchiolitis incidence for years 2016–2019 was well captured by the model ($R^2$ = 0,84). Incidence of bronchiolitis decreased during the first lockdown (Fig 3). In autumn, where incidence was seen to increase in the past, the second lockdown occurred and a large decrease with respect to expected levels was noted. Overall, incidence dropped by 79% from the expected incidence during the first lockdown and by 78% during the second one. The search query volume for "bronchiolitis" showed a similar plunge during the second lockdown.

As observed in the chickenpox and AD relative reduction, some regional variation can be observed in the bronchiolitis relative reduction, ranged from 45% to 75% during the first lockdown (S5 Fig). There was a positive correlation between the relative reduction and the number

**Table 2. Difference between observed and expected cumulative incidence and number of GP's consultation in France metropolitan.**

| Indicator | Period | Cumulative incidence (standard error) | Cumulative expected incidence (standard error) | Reduction (%) |
|---|---|---|---|---|
| Acute diarrhea (AD) | Pre-lockdown | 1901 (74) | 1849 (351) | -3 |
| | Lockdown | 252 (24) | 764 (144) | 67 |
| | Post-lockdown | 386 (31) | 610 (115) | 37 |
| | Summer vacation | 422 (37) | 643 (121) | 34 |
| | Autumn | 330 (25) | 622 (117) | 47 |
| | Lockdown 2 | 248 (20) | 681 (129) | 64 |
| Acute respiratory infection (ARI) | Pre-lockdown | 5965 (73) | 6884 (2910) | 13 |
| | Lockdown | 921 (23) | 1804 (757) | 49 |
| | Post-lockdown | 571 (19) | 1062 (446) | 46 |
| | Summer vacation | 648 (22) | 1092 (458) | 41 |
| | Autumn | 1007 (22) | 1166 (489) | 14 |
| | Lockdown 2 | 691 (17) | 1519 (638) | 55 |
| Chickenpox | Pre-lockdown | 165 (22) | 245 (150) | 33 |
| | Lockdown | 23 (6) | 235 (143) | 90 |
| | Post-lockdown | 6 (4) | 198 (121) | 97 |
| | Summer vacation | 16 (8) | 57 (35) | 72 |
| | Autumn | 22 (7) | 26 (16) | 18 |
| | Lockdown 2 | 34(7) | 41 (25) | 18 |
| Bronchiolitis | Pre-lockdown | 124 (11) | 157 (64) | 21 |
| | Lockdown | 12 (3) | 57 (23) | 79 |
| | Post-lockdown | 11 (3) | 33 (13) | 67 |
| | Summer vacation | 15 (3) | 24 (10) | 38 |
| | Autumn | 28 (4) | 37 (15) | 24 |
| | Lockdown 2 | 16 (3) | 73 (30) | 78 |
| | | Cumulative number of consultation per GP | Cumulative expected Number of consultation per GP | Reduction (%) |
| Consultation in primary care | Pre-lockdown | 1306 | 1326 | 1 |
| | Lockdown | 581 | 803 | 28 |
| | Post-lockdown | 753 | 766 | 2 |
| | Summer vacation | 746 | 806 | 8 |
| | Autumn | 665 | 646 | -3 |
| | Lockdown 2 | 512 | 541 | 5 |

of hospitalizations but not statistically significant (Pearson's correlation coefficient with number of hospitalizations r = 0,34).

## ARI and consultations

The incidence rate of ARI was lower than the preceding years (Table 2). The decrease was smaller during the first lockdown (49%). Incidence was close to the expected after the Summer Holidays, but decreased again during the second lockdown. The regional variation observed for the relative reduction in ARI incidence ranged from 20% to 50% during the first lockdown (S6 Fig). We found that, in contrast to AD incidence reduction, regions where COVID-19 circulation had been the largest in the first wave experienced the lowest reduction in ARI incidence (Pearson's correlation coefficient with number of hospitalizations r = -0,74).

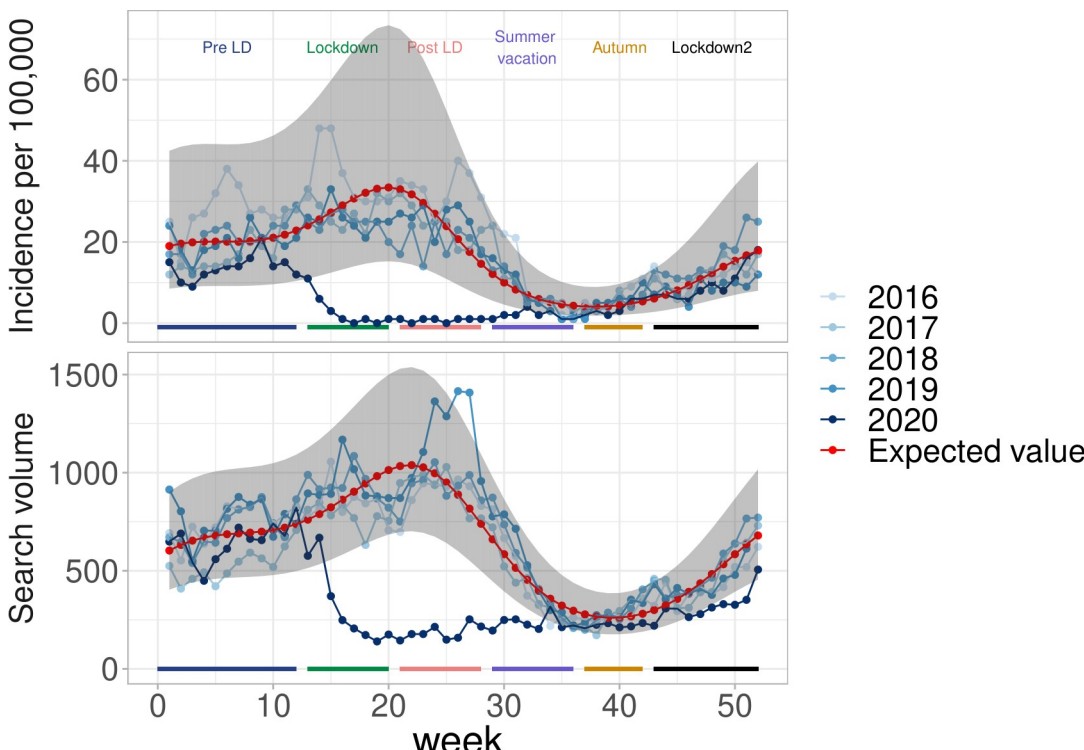

**Fig 2. Chickenpox in France during the COVID-19 pandemic.** (top) Incidence in general practice and (bottom) search query volume.

We found that in the year 2020, patients presented with ARI symptoms who underwent a PCR test were most likely infected with SARS-CoV-2 or human metapneumoviruses, (S4 Fig) in sharp contrast with year 2019 where they were likely infected with influenza. In 2020, only two cases of influenza and one case of RSV were detected (Fig 4).

**Consultation volume in the general population.**   In France, the number of consultations with GPs decreased from the expected during the first two weeks of the first lockdown, but started increasing again in the 5th week of the first lockdown. At the end of the first lockdown period, the number of consultations was again close to the expected, and has remained so ever since (Fig 5). The participatory surveillance system GrippeNet.fr, where participants indicated whether they contacted a GP in case of symptoms, showed that consultations with GPs for ARI did not change importantly over the first half of year 2020: in the pre-lockdown period, 32% of ARI patients consulted with a GP for that episode, it was 51% and 46% in the first lockdown and post-lockdown phase. An increase in the number of teleconsultations with GPs was observed over time (S1 Fig).

## Discussion

Following the introduction of SARS-CoV-2 in France, a number of measures have been adopted to reduce transmission, culminating with a first national lockdown in March-May 2020. The main goal of these preventive measures was to slow down the spread of the SARS-CoV-2 by reducing the number of contacts between individuals. We found that during the same period of time, the circulation of other common communicable diseases presented major decreases from the expected. The relative decrease in incidence for AD (67%) and

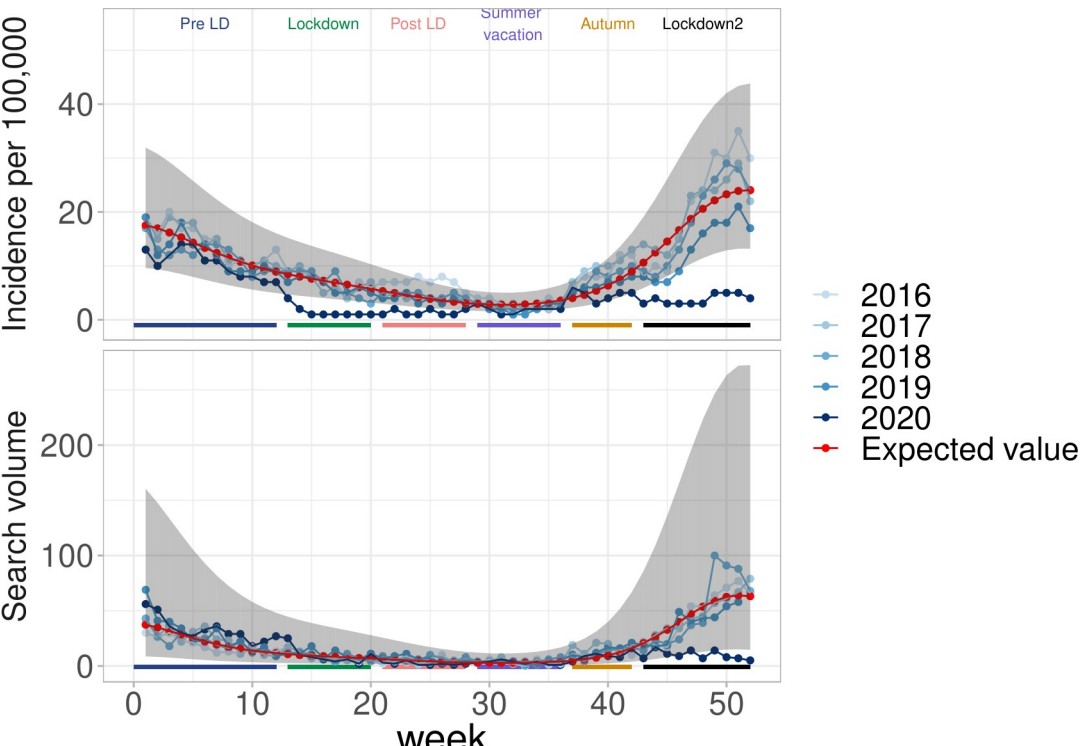

**Fig 3. Bronchiolitis in France during the COVID-19 pandemic.** (top) Incidence in general practice (IQVIA) and (bottom) search query volume.

chickenpox (90%) during the lockdown period were of the same magnitude as the reduction of the reproduction ratio of COVID-19 (77%) [2].

Measuring the impact of interventions on the circulation of a communicable disease is difficult, for want of a clear reference situation. Here, we relied on data collected routinely by two surveillance systems in general practice, the Sentinelles network [21] and the EMR database from IQVIA, in the four years preceding 2020 and throughout year 2020. Both systems collect data directly from GPs, using standardized clinical case definitions for the Sentinelles network or International Classification of Diseases (ICD) coding for the EMR database. The two systems collect data independently. Incidence computed in the Sentinelles network gives an accurate picture of the epidemiology of common communicable diseases in France: for example, there is very good correlation between the Sentinelles AD incidence and the actual volume of drugs prescribed for AD episodes (rehydration and anti-vomiting) and with the number of emergency departments visits for AD [18], and the Sentinelles incidence of chickenpox also corresponds well with the epidemiology of the disease [17, 22], with rates following the pattern of school holidays as in most countries [23]. Both AD and chickenpox show strong seasonal patterns, which are however remarkably consistent from one year to the next. This provided a unique opportunity to compute the expected incidence of these diseases and provide reference numbers for comparison with the observed epidemiology during the COVID-19 period. We adopted a periodic regression analysis to capture the strong seasonal patterns of incidence and found that the fitted models and expected intervals were well calibrated. In the two first months of 2020, before intense circulation of COVID-19, the observed incidence of ARI, AD, chickenpox and bronchiolitis were well within the expected ranges.

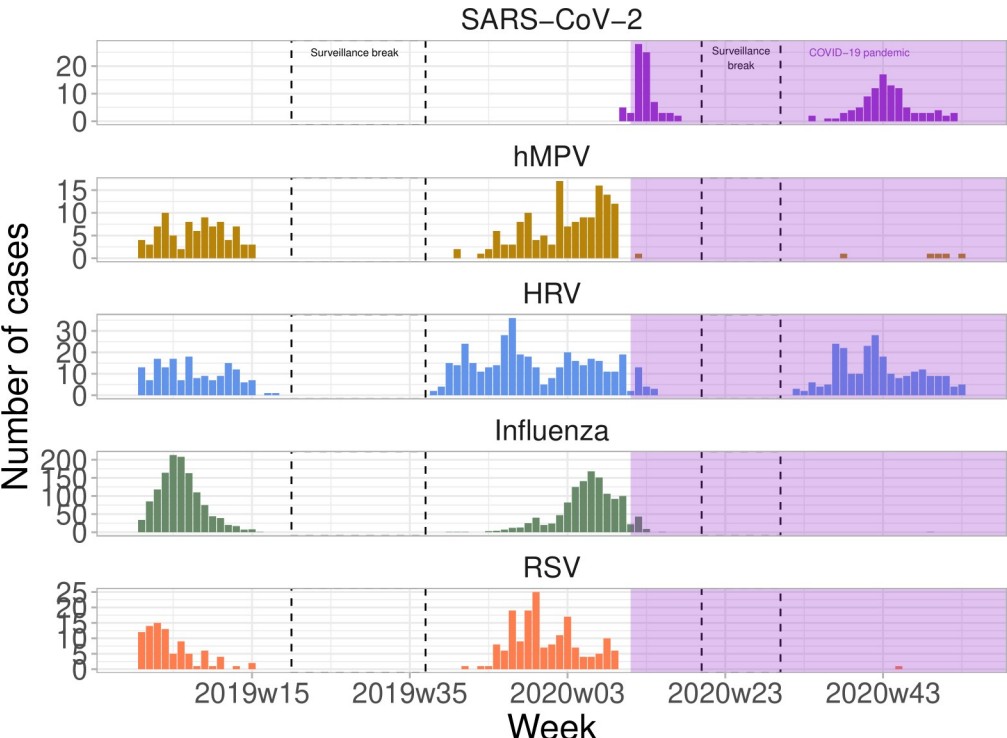

**Fig 4. Number of viruses detected by virological confirmation in ARI consultation.**

Barrier measures have been adopted in France since the beginning of March 2020 [24], culminating with national lockdowns in two instances. These measures are expected to have an impact on the circulation of most directly and indirectly transmitted diseases. Swabs obtained

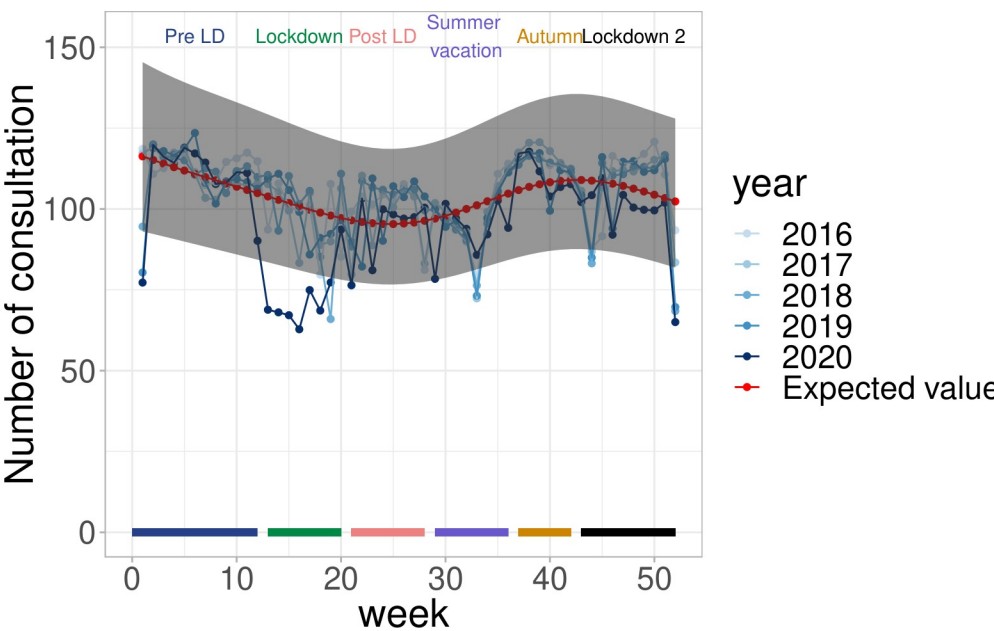

**Fig 5. Number of general practitioner consultation in France during COVID-19 pandemic, IQVIA.**

from patients in the Sentinelles network show that the viruses circulating during the COVID-19 pandemic in ARI patients, were different than in the previous season, with neither influenza nor RSV in 2020 when these viruses had been the most frequent in 2019. We indeed observed that the incidence of ARI and bronchiolitis was less than expected during year 2020, as were the incidence of AD and chickenpox. The periods where restrictions were the strongest in the population corresponded with the largest reduction in incidence. For example, the reduction of AD was by 2/3 during the lockdown periods, but closer to 1/3 in the other periods. Childhood diseases almost vanished. Chickenpox, which circulates the whole year round in France, was reduced close to 0 during the periods where schools were closed. It increased again from September 2020 as children resumed going to school, although with levels less than the expected. Bronchiolitis, which affects mostly pre-school children, was almost absent during its period of main activity (October/November). For AD, which affects all age classes, we found that the reduction in incidence was largely in the children, less in adults and almost zero among the elderly. This is in line with the distribution of the number of contacts according to age [25]: Social distancing measures are likely to have had a larger impact on the young, who have a lot of contacts, than on others age classes. Interestingly, we found that regions where the reduction in incidence for AD was the largest were those where the COVID-19 epidemics had led to more hospitalizations. Population in the most affected regions may have had more incentive on adopting precautionary measures and respecting recommendations. Previous work found indeed that drops in mobility were larger in the regions mostly hit by the pandemic in the first wave [26]. Also, survey data indicates that adherence to preventive measures increases with personal experience of the disease, for example with a relative affected by COVID-19 [24].

The decrease in incidence as measured by the indicators in general practice could reflect a general reduction in consultations with general practitioners. Indeed, the number of visits decreased by approximately 29% in the first 5 weeks of the first lockdown, as measured by the IQVIA EMR database. This decrease was temporary and the consultation rate almost regained its expected level after 5 weeks, contrary to incidence of the monitored diseases that was affected during the whole period. The decrease in GP visits can have several reasons. First, patients may have avoided consultations despite the recommendations of national health authorities. Indeed, french GPs reported a reduction in activity during the first lockdown, with a decrease in consultations for chronic diseases and a rise in those for mental health issues [27]. A reduction in consultations for acute conditions such as ARI, AD, chickenpox or bronchiolitis may have taken place as well, although participants to the GrippeNet.fr participatory surveillance system reported consulting/teleconsulting with a GP in case of ARI as often before, during and after the lockdown (S1 Fig). Second, teleconsultations may not have been properly recorded in surveillance systems as they were rare before 2020. Indeed, teleconsultations increased from less than 40,000 in February 2020 to more than 2 million in March 2020 [28]. In all instances, consultations initially decreased less than disease incidence during the first lockdown: it was 30% when the reduction in disease incidence was twice as much. Furthermore, the reduction in consultations was less marked during the second lockdown, but the reduction in incidence of AD, bronchiolitis and ARI were almost the same than in the first lockdown. Consultations for acute respiratory infections, acute diarrhea, chickenpox and bronchiolitis correspond to less than 10% of the 200M GP consultations each year. GPs report that after the initial drop in visits during the first lockdown, there was an increase in consultations for COVID-19 and for mental health issues [29]. This increase may explain why the overall number of consultations regained expected levels after the first lockdown, when disease incidence did not. It is therefore unlikely that all reductions in disease incidence are a consequence of less consultations. The change similar to incidence seen in alternative sources of

information also supports a real decrease. For example, search query volume has previously been correlated with the incidence of AD or chickenpox [14]. Here, the patterns in search query volume paralleled that of incidence, with large troughs during the time of lockdown. We also found that drugs typically prescribed in AD episodes were less dispensed in pharmacies during the lockdown. Last, the proportion of bronchiolitis visits in hospital emergency departments visits for the children under two years old remained far below the previous year: in 2020–2021, this proportion was less than 5% when it was around 15% in 2018–2019 [30].

A reduction of the circulation of common communicable disease has been observed in other countries. In Australia, strict control measures have been implemented such as working from home, closing national borders, limiting on social gatherings and school closure (in the Victoria's state). Right after the travel and work restrictions were adopted, the circulation of Influenza viruses and RSV (respiratory syncytial virus) dropped. During the same period, ILI consultations in 2020 remained far below the historical average [31]. Influenza detection was reduced by 99.4% and the RSV detection by 98% from the previous year, even though the number of swabs tested was higher than the previous year [32]. In Finland, restrictive measures to social distance were also implemented: school and national border closure, as well as work from home were recommended, and residents over 70 years were ordered to stay at home. The 2020 influenza season lasted only eight weeks (from peak to no case) compared to 13–20 weeks in the 4 previous years. A rapid decline was also observed in RSV circulation [33]. In Hong Kong, influenza transmission in the community dropped substantially after the implementation of non-pharmaceutical interventions in late January, with a reduction of 44% (95% CI 34–53%) [34]. This was observed while large adoption rates were recorded in the population for the use of preventive measures (e.g. use of mask and avoidance of crowded spaces). These changes may however not be present in all countries, depending on the circulation of SARS-CoV-2 and the adoption of mitigation measures.

The reduction in incidence for these common diseases finally raises the question of an eventual catch-up. It was predicted that substantial RSV outbreaks could occur in the coming years [35, 36]. AD, which is mainly caused by rotavirus, astroviruses and norovirus in the winter [37], could lead to similar changes. For chickenpox, where 90% of the french children are infected by age 10, more than half of the expected 600000 yearly cases did not occur in 2020. While the catch-up may be rapid in the coming years, a larger fraction may reach adulthood as non-immune and require vaccination to prevent complications.

Our study has the following limitations. First, incidence data is based on a case definition using clinical information only. Virological information was available for common respiratory viruses, but not for AD and chickenpox. Second, while our study consistently shows that the incidence of common infectious diseases was reduced during time periods where the strongest measures for COVID-19 were implemented, it does not allow causal interpretation. Third, the usage of search query terms for infectious diseases may have changed during the pandemic, reflecting anxiety more than disease. The query terms are however very specific french words for chickenpox, diarrhea and bronchiolitis. While the trends in search terms were similar to that of actual incidences, it could be due to changes in health seeking over the internet rather than actual changes in health conditions. Fourth, there was a global reduction in the overall number of GP consultations, especially at the beginning of the first lockdown. This likely contributed to the observed decrease in incidence. Finally, AD is known to be caused by multiple pathogens, viruses or bacteria [37, 38]. An overall decrease does not imply that all pathogens were equally affected.

Standard precautionary measures and social distancing in place today in many countries may have sufficient effect, at least temporarily, to hinder transmission of common winter

viruses. Surveillance systems will be of interest in monitoring disease activity in the years to come.

## Supporting information

**S1 Fig. Percentage of consultation for ILI syndrom in Grippenet cohort in France during COVID-19 pandemic.**
(TIF)

**S2 Fig. Acute diarrhea in France during the COVID-19 pandemic, incidence rate by age in general practice.**
(TIF)

**S3 Fig. Cumulated incidence of COVID-19 hospitalisations over the first lockdown period according to relative reduction in acute diarrhea incidence by administrative regions in France (r = 0,59).**
(TIF)

**S4 Fig. Cumulated incidence of COVID-19 hospitalisations over the first lockdown period according to relative reduction in chickenpox by administrative regions in France (r = -0,05).**
(TIF)

**S5 Fig. Cumulated incidence of COVID-19 hospitalisations over the first lockdown period according to relative reduction in bronchiolitis incidence by administrative regions in France (r = 0,34).**
(TIF)

**S6 Fig. Cumulated incidence of COVID-19 hospitalisations over the first lockdown period according to relative reduction in acute respiratory infection incidence by administrative regions in France (r = -0,74).**
(TIF)

## Author Contributions

**Conceptualization:** Pierre-Yves Boëlle.

**Data curation:** Titouan Launay, Clément Turbelin, Caroline Guerrisi, Isabelle Bardoulat, Magali Lemaître.

**Formal analysis:** Titouan Launay.

**Methodology:** Titouan Launay, Cécile Souty, Ana-Maria Vilcu, Pierre-Yves Boëlle.

**Supervision:** Pierre-Yves Boëlle.

**Writing – original draft:** Titouan Launay, Pierre-Yves Boëlle.

**Writing – review & editing:** Cécile Souty, Ana-Maria Vilcu, Clément Turbelin, Thierry Blanchon, Caroline Guerrisi, Thomas Hanslik, Vittoria Colizza, Isabelle Bardoulat, Magali Lemaître.

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
