## [Decision Letter · Decision Letter 0]

24 Jun 2021

PONE-D-21-16521

Common communicable diseases during the COVID-19 pandemic in the general population in France

PLOS ONE

Dear Dr. Launay,

Thank you for submitting your manuscript to PLOS ONE. After careful consideration, we feel that it has merit but does not fully meet PLOS ONE’s publication criteria as it currently stands. Therefore, we invite you to submit a revised version of the manuscript that addresses the points raised during the review process.

Please respond to the reviewer comments on a point-by-point basis and revise the manuscript accordingly.  In addition, please vet the manuscript writing thoroughly for grammar and syntax.

We look forward to receiving your revised manuscript.

Kind regards,

Jeffrey Shaman

Academic Editor

PLOS ONE

Journal Requirements:

We note that one or more of the authors are employed by a commercial company: "IQVIA"

3. We note you have included a table to which you do not refer in the text of your manuscript. Please ensure that you refer to Table 2 in your text; if accepted, production will need this reference to link the reader to the Table.

Reviewers' comments:

Reviewer's Responses to Questions

**Comments to the Author**

1. Is the manuscript technically sound, and do the data support the conclusions?

Reviewer #1: Yes

Reviewer #2: Yes

2. Has the statistical analysis been performed appropriately and rigorously? 

Reviewer #1: Yes

Reviewer #2: Yes

3. Have the authors made all data underlying the findings in their manuscript fully available?

Reviewer #1: Yes

Reviewer #2: No

4. Is the manuscript presented in an intelligible fashion and written in standard English?

Reviewer #1: Yes

Reviewer #2: Yes

5. Review Comments to the Author

Reviewer #1: This manuscript analyzed syndromic, prescription and online search data to examine patterns of some common groups of ailments pre and during the pandemic to see if trends changed. The 2020 year was divided up by when measures were in place, and history data was used as the comparator. In general, diarrheal, bronchial and respiratory diseases and chicken pox all declined; the effect was greatest for the first big lockdown. This first lockdown also had a drop in GP consultations, suggesting some of the drop could have been the result of changes in health seeking behaviors. The ongoing decline in illness despite the return of GP consultations lend support that the mitigation effects interrupted disease transmission for other viral diseases.

Specific comments

Line 1. I would add “in France” to the title.

Line 38. I am not sure what you mean by “…in line with the extent of implemented measures.” It seems like this was unprecedented territory so not sure how this was assessed. I realize it may be explained in the paper but probably best to reword for the abstract.

Line 39. Maybe I missed it, but I didn’t see the age specific results anywhere in the paper.

Line 43. I think you “imply” or “indirectly show” that other viruses decreased in circulation. You don’t actually present any viral data; also, this is an ecological analysis, so you are really only showing correlation and not causation.

Line 54. Do you mean “remained closed” or just “closed?” If this was the first closing, then you should reword.

Line 59-60. Awkward phrasing, please revise.

Line 68. Change to “decreased.”

Line 76. Somewhere in the methods you need to indicate what the public was told to do for routine health care. Were people told not to seek care? Were they told to call instead of seek care? Were all non-urgent medical issues triaged to a central location that would no be picked up routine systems. The issue is that you need to be able to tease out the role of changing in health seeking behavior vs. a true decrease in actual illnesses. Your data on GP consultations also get at this issue. Also, did you consider having a control illness that might not have expected to have changes due to the social distancing measures put in place? This would be one way to tease out a true decline vs. one due to changes in health seeking behaviors. You allude to this (line 300) but I would like it might be easy to add to the analysis.

Line 78. Please indicate somewhere in the methods whether any of these data sources include laboratory results.

Line 159, line 190-191, line 200. It says “…Error! Reference source not found.”

Line 245-246. When you say “indirect evidence of the effect of mitigation measures” do you mean on SARS-CoV-2? You need to be clear. I think these findings are indirect evidence of reducing the burden of illness from the diseases you were analyzing (it’s indirect because it’s ecological and you don’t have any lab data). You did not really analyze any COVID data in this paper, so I think that is a bit off topic. This is again stated in line 337 and I would revise it here too. If you wanted to look at the effect on COVID then you should include those data as well and make that a central objective up front.

Line 286. I think you mean “adherence” instead of “adhesion.”

Line 339. I might add “at least temporarily” since we don’t really know how long the effects will last, and in fact some viruses are returning.

I think you need a limitations paragraph. A few limitations I can think of are 1) ecological analysis, 2) no laboratory data, 3) some of the illnesses and very broad groups that may be affected differently by the mitigation measures (e.g., ARI could include COVID-19 but it also might include rhino/enteroviruses which have been shown to be less impacted by the measures).

Reviewer #2: This manuscript uses multiple lines of evidence (search engine query volume, general practitioner incidence reports, and drug sales) to document a reduction in diarrhea, chickenpox, bronchiolitis, and acute respiratory infections during France’s COVID-19 lockdowns. Although this finding is not entirely surprising, it is important that these major epidemiological shifts be documented and studied. The statistical approach employed by the authors is sound and well-reasoned, and the article is well written. Please see the suggestions below:

1. There should be a discussion of study limitations. For instance, one limitation that immediately stands out is the fact that most measures of disease incidence included in the study are indirect (drug sales, and search engine volume). Physician reported incidence is also in a way indirect as these only survey a limited number of physicians and may be subject to bias. In the absence of laboratory data showing decreased isolation of specific pathogens in a well sampled population, the conclusions should be interpreted with caution and these limitations mentioned.

2. Over the counter drug purchases are an imperfect sentinel for disease as drugs may have been purchased in anticipation of disease or in response to media reports etc. Likewise, search terms may reflect anxiety regarding COVID and not necessarily reflect disease. Did the authors take this into account when selecting their search terms?

3. Regional variation for diarrhea, chickenpox, bronchiolitis, and acute respiratory infections are referenced frequently but these data are not shown. Suggest showing these regional data (perhaps in supplemental table format) and referencing these data when these variations are mentioned.

4. It is somewhat surprising that symptom search terms did not increase during periods of heavy COVID transmission, given that many individuals likely had COVID or were researching COVID. The potential contribution of COVID disease or concern about COVID disease on the data should be discussed.

5. The authors should discuss the fact that their results may not be applicable to other locations, given potentially unique characteristics of the COVID-19 epidemic in France or differences in country response.

6. Did food consumption patterns change significantly during lockdown? It is possible that people still bought food from the same stores, etc. The authors should discuss hypotheses for why diarrhea rates declined. Fecal oral transmission via food likely persisted to some degree, whereas person-to-person contact (a key route of viral gastroenteritis transmission) likely declined. As viral, bacterial, or parasitic causes are not distinguished by the authors methodology, it is possible (even likely) that some pathogens declined in incidence while others remained the same. This should be discussed.

7. The authors show that clinic visits dropped during the first lockdown but were otherwise at roughly the same levels during COVID as prior years. This would seem to contradict the sharp decreases in incidence of common infections (diarrhea, acute respiratory infections, etc), as estimated by drug sales, search queries, and general practitioner incidence reports. These infections presumably account for a large share of clinic visits. How do the authors explain this?

A few specific minor suggestions below:

Line 57: please state when the second lock-down ended.

Line 67: please provide a reference for the statement regarding mask wearing rates.

Line 86: “at least three daily watery or nearly so stools” -typo?

Lines 159, 190, and 220: references not loaded correctly

Lines 223-226: this sentence does not make sense. Should the second “AD” be “ARI”?

6. PLOS authors have the option to publish the peer review history of their article (what does this mean?). If published, this will include your full peer review and any attached files.

Reviewer #1: No

Reviewer #2: No

---

## [Author Response · Author response to Decision Letter 0]

25 Aug 2021

Reviewer 1 :

- Line 1. I would add “in France” to the title. 

We changed the title to : Common communicable diseases during the COVID-19 pandemic in the general population, in France.

- Line 38. I am not sure what you mean by “…in line with the extent of implemented measures.” It seems like this was unprecedented territory so not sure how this was assessed. I realize it may be explained in the paper but probably best to reword for the abstract.

We changed to : “Reductions with respect to the expected incidence reflected the strength of implemented measures.”

- Line 39. Maybe I missed it, but I didn’t see the age specific results anywhere in the paper. 

Age specific results were only described in the Acute diarrhea section (line 172-173) and in figure S2, as bronchiolitis and chickenpox are only seen in the young.

- Line 43. I think you “imply” or “indirectly show” that other viruses decreased in circulation. You don’t actually present any viral data; also, this is an ecological analysis, so you are really only showing correlation and not causation.

We changed to : “In primary care, the large decrease in reported gastroenteritis, chickenpox or bronchiolitis observed during the period where many barrier measures were implemented imply that the circulation of common viruses was reduced and informs on the overall effect of these measures.”

- Line 54. Do you mean “remained closed” or just “closed?” If this was the first closing, then you should reword.

We changed to : ”A first national lockdown was adopted from March 17th to May 11th, during which mobility was restricted, schools, public services and shops were closed and telework widely adopted.”

- Line 59-60. Awkward phrasing, please revise.

We changed to : “These transmission routes are the same for other respiratory viruses. Standard precautionary measures should therefore limit transmission for all these diseases.”

- Line 68. Change to “decreased.”

We changed to “decreased”.

- Line 76. Somewhere in the methods you need to indicate what the public was told to do for routine health care. Were people told not to seek care? Were they told to call instead of seek care? Were all non-urgent medical issues triaged to a central location that would no be picked up routine systems. 

During the pandemic, the general public was encouraged to keep consulting with their GPs as normal.(https://solidarites-sante.gouv.fr/actualites/presse/communiques-de-presse/article/covid-19-et-continuite-des-soins-continuer-de-se-soigner-un-imperatif-de-sante) No triage system was set up that would explain a drop in consultations. Importantly, teleconsultation is reimbursed as normal visits.

This is mentioned at the beginning of the Methods: 

“During the pandemic, the general population was encouraged to consult with their GPs as usual. No triage system was put into place. Importantly, teleconsultations by phone or internet were reimbursed as normal visits.”

The issue is that you need to be able to tease out the role of changing in health seeking behavior vs. a true decrease in actual illnesses. Your data on GP consultations also get at this issue. Also, did you consider having a control illness that might not have expected to have changes due to the social distancing measures put in place? This would be one way to tease out a true decline vs. one due to changes in health seeking behaviors. You allude to this (line 300) but I would like it might be easy to add to the analysis.

We only had data on infectious diseases, so that it was not possible to use other conditions for control. Regarding the drop in consultations with GPs, it was large for 5 weeks after the first lockdown started, but consultations regained their expected level afterwards, contrary to incidence for the monitored diseases. In the discussion, we listed several explanations that contributed to the initial drop, and we acknowledged that renunciation to care was one of them. We reinforced in the discussion that part of the observed decrease could be due to change in health seeking behavior, quoting official sources on the role of GPs during the first lockdown.

We changed the discussion to : 

“This decrease was temporary and the consultation rate almost regained its expected level after 5 weeks, contrary to incidence of the monitored diseases that was affected during the whole period. The decrease in GP visits can have several reasons. First, patients may have avoided consultations despite the recommendations of national health authorities. Indeed, french GPs reported a reduction in activity during the first lockdown, with a decrease in consultations for chronic diseases and a rise in those for mental health issues[26]. A reduction in consultations for acute conditions such as ARI, AD, chickenpox or bronchiolitis may have taken place as well, although participants to the GrippeNet.fr participatory surveillance system reported consulting/teleconsulting with a GP in case of ARI as often before, during and after the lockdown (Figure S1).”

and 

“Our study has the following limitations. First, incidence data is based on a case definition using clinical information only. Virological information was available for common respiratory viruses, but not for AD and chickenpox. Second, while our study consistently shows that the incidence of common infectious diseases was reduced during time periods where the strongest measures for COVID-19 were implemented, it does not allow causal interpretation. Third, the usage of search query terms for infectious diseases may have changed during the pandemic, reflecting anxiety more than disease. The query terms are however very specific french words for chickenpox, diarrhea and bronchiolitis. While the trends in search terms were similar to that of actual incidences, it could be due to changes in health seeking over the internet rather than actual changes in health conditions. Fourth, there was a global reduction in the overall number of GP consultations, especially at the beginning of the first lockdown. This likely contributed to the observed decrease in incidence. Finally, AD is known to be caused by multiple pathogens, viruses or bacteria[35,36]. An overall decrease does not imply that all pathogens were equally affected.” 

- Line 78. Please indicate somewhere in the methods whether any of these data sources include laboratory results.

We only have virological data for a sample of patients presenting with ARI. Indeed in the Sentinelles network, a random subset of patients presenting with ARI symptoms undergo nasopharyngeal swabs followed by RT-PCR to test for 4 viruses (influenza, human metapneumovirus, respiratory syncytial virus and rhinovirus). SARS-CoV-2 was added in March 2020.

We now report the changes in viruses isolated before the COVID-19 pandemic and after. 

We added a sentence in the methods : 

“Incidence in the IQVIA and Sentinelles network are based on clinical diagnoses. Additionally, in the Sentinelles network, a random subset of patients presenting with ARI symptoms undergo nasopharyngeal swabs followed by RT-PCR to test for 4 viruses (influenza, human metapneumovirus, respiratory syncytial virus and rhinovirus), and for SARS-CoV-2 since March 2020 [11].”

And in the results : 

“We found that in the year 2020, patients presented with ARI symptoms who underwent a PCR test were most likely infected with SARS-CoV-2 or human metapneumoviruses, (Figure S4) in sharp contrast with year 2019 where they were likely infected with influenza. In 2020, only two cases of influenza and one case of RSV were detected (Fig 4).”

And in the discussion : 

“Swabs obtained from patients in the Sentinelles network show that the viruses circulating during the COVID-19 pandemic in ARI patients, were different than in the previous season, with neither influenza nor RSV in 2020 when these viruses had been the most frequent in 2019.”

- Line 245-246. When you say “indirect evidence of the effect of mitigation measures” do you mean on SARS-CoV-2? You need to be clear. I think these findings are indirect evidence of reducing the burden of illness from the diseases you were analyzing (it’s indirect because it’s ecological and you don’t have any lab data). You did not really analyze any COVID data in this paper, so I think that is a bit off topic. This is again stated in line 337 and I would revise it here too. If you wanted to look at the effect on COVID then you should include those data as well and make that a central objective up front.

We now removed references to the efficiency of control measures for COVID-19.

We changed line 245-246 to : “We found that during the same period of time, the circulation of other common communicable diseases presented major decreases from the expected.” 

- Line 286. I think you mean “adherence” instead of “adhesion.”

We changed to adherence : “Also, survey data indicates that adherence to preventive measures increases with personal experience of the disease, for example with a relative affected by COVID-19[23].”

- Line 339. I might add “at least temporarily” since we don’t really know how long the effects will last, and in fact some viruses are returning.

I think you need a limitations paragraph. A few limitations I can think of are 1) ecological analysis, 2) no laboratory data, 3) some of the illnesses and very broad groups that may be affected differently by the mitigation measures (e.g., ARI could include COVID-19 but it also might include rhino/enteroviruses which have been shown to be less impacted by the measures).

We added a limitation paragraph : 

“Our study has the following limitations. First, incidence data is based on a case definition using clinical information only. Virological information was available for common respiratory viruses, but not for AD and chickenpox. Second, while our study consistently shows that the incidence of common infectious diseases was reduced during time periods where the strongest measures for COVID-19 were implemented, it does not allow causal interpretation. Third, the usage of search query terms for infectious diseases may have changed during the pandemic, reflecting anxiety more than disease. The query terms are however very specific french words for chickenpox, diarrhea and bronchiolitis. While the trends in search terms were similar to that of actual incidences, it could be due to changes in health seeking over the internet rather than actual changes in health conditions. Fourth, there was a global reduction in the overall number of GP consultations, especially at the beginning of the first lockdown. This likely contributed to the observed decrease in incidence. Finally, AD is known to be caused by multiple pathogens, viruses or bacteria[35,36]. An overall decrease does not imply that all pathogens were equally affected. “

 Reviewer 2 :

- 1. There should be a discussion of study limitations. For instance, one limitation that immediately stands out is the fact that most measures of disease incidence included in the study are indirect (drug sales, and search engine volume). Physician reported incidence is also in a way indirect as these only survey a limited number of physicians and may be subject to bias. In the absence of laboratory data showing decreased isolation of specific pathogens in a well sampled population, the conclusions should be interpreted with caution and these limitations mentioned.

We added a limitation in a specific paragraph.

“Our study has the following limitations. First, incidence data is based on a case definition using clinical information only. Virological information was available for common respiratory viruses, but not for AD and chickenpox.“

We now report the changes in viruses isolated before the COVID-19 pandemic and after. 

We added a sentence in the methods : 

“Incidence in the IQVIA and Sentinelles network are based on clinical diagnoses. Additionally, in the Sentinelles network, a random subset of patients presenting with ARI symptoms undergo nasopharyngeal swabs followed by RT-PCR to test for 4 viruses (influenza, human metapneumovirus, respiratory syncytial virus and rhinovirus), and for SARS-CoV-2 since March 2020 [11].”

And in the results : 

“We found that in the year 2020, patients presented with ARI symptoms who underwent a PCR test were most likely infected with SARS-CoV-2 or human metapneumoviruses, (Figure S4) in sharp contrast with year 2019 where they were likely infected with influenza. In 2020, only two cases of influenza and one case of RSV were detected (Fig 5).”

And in the discussion : 

“Swabs obtained from patients in the Sentinelles network show that the viruses circulating during the COVID-19 pandemic in ARI patients, were different than in the previous season, with neither influenza nor RSV in 2020 when these viruses had been the most frequent in 2019.”

- 2. Over the counter drug purchases are an imperfect sentinel for disease as drugs may have been purchased in anticipation of disease or in response to media reports etc. Likewise, search terms may reflect anxiety regarding COVID and not necessarily reflect disease. Did the authors take this into account when selecting their search terms?

The search terms we used are really the french words used to describe AD and chickenpox. These terms have been shown to accurately reflect incidence of the two diseases[1]. Since 2009 the same keywords for Acute diarrhea and chickenpox are used in routine to compare the incidence and the search trend and we kept the same method for 2020. 

The sanitary situation due to the COVID-19 pandemic, may have an impact in the search trends, nevertheless the correlation with the incidence remained high in 2020 suggesting that the reduction in search trends is a consequence to the reduction of viruses circulation. 

We added a limitation in a specific paragraph.

“Third, the usage of search query terms for infectious diseases may have changed during the pandemic, reflecting anxiety more than disease. The query terms are however very specific french words for chickenpox, diarrhea and bronchiolitis. While the trends in search terms were similar to that of actual incidences, it could be due to changes in health seeking over the internet rather than actual changes in health conditions.”

- 3. Regional variation for diarrhea, chickenpox, bronchiolitis, and acute respiratory infections are referenced frequently but these data are not shown. Suggest showing these regional data (perhaps in supplemental table format) and referencing these data when these variations are mentioned.

The regional variation in Diarrhea incidence are presented in the figure S2 in the supporting information. We added the corresponding figure for chickenpox, bronchiolitis and acute respiratory infection.

- 4. It is somewhat surprising that symptom search terms did not increase during periods of heavy COVID transmission, given that many individuals likely had COVID or were researching COVID. The potential contribution of COVID disease or concern about COVID disease on the data should be discussed. 

The search terms “varicelle” (chickenpox), “bronchiolite” (bronchiolitis) correspond to the disease name and not symptoms. These are unlikely to be used in the place of COVID-19. The term “gastro” (acute diarrhea) could indeed be a symptom of COVID-19. 

 We included a limitations paragraph where this was discussed.

“Third, the usage of search query terms for infectious diseases may have changed during the pandemic, reflecting anxiety more than disease. The query terms are however very specific french words for chickenpox, diarrhea and bronchiolitis. While the trends in search terms were similar to that of actual incidences, it could be due to changes in health seeking over the internet rather than actual changes in health conditions.”

- 5. The authors should discuss the fact that their results may not be applicable to other locations, given potentially unique characteristics of the COVID-19 epidemic in France or differences in country response.

We added at the end of the discussion :

“These changes may however not be present in all countries, depending on the circulation of SARS-CoV-2 and the adoption of mitigation measures.”

- 6. Did food consumption patterns change significantly during lockdown? It is possible that people still bought food from the same stores, etc. The authors should discuss hypotheses for why diarrhea rates declined. Fecal oral transmission via food likely persisted to some degree, whereas person-to-person contact (a key route of viral gastroenteritis transmission) likely declined. As viral, bacterial, or parasitic causes are not distinguished by the authors methodology, it is possible (even likely) that some pathogens declined in incidence while others remained the same. This should be discussed.

 We included a limitations paragraph where this was discussed.

“Finally, AD is known to be caused by multiple pathogens, viruses or bacteria[35,36]. An overall decrease does not imply that all pathogens were equally affected.”

- 7. The authors show that clinic visits dropped during the first lockdown but were otherwise at roughly the same levels during COVID as prior years. This would seem to contradict the sharp decreases in incidence of common infections (diarrhea, acute respiratory infections, etc), as estimated by drug sales, search queries, and general practitioner incidence reports. These infections presumably account for a large share of clinic visits. How do the authors explain this?

The general practitioner were in the front line during the COVID-19 pandemic. The public was encouraged to keep consulting and teleconsultations was fully reimbursed. Furthermore, the consultations for mental health issue increase, especially during the first lockdown. 

Only 8% of the general practitioner reported a 10 hours per week decrease in their volume of work after the first lockdown. During the second lockdown this ratio fall to 3%. According to 54% of the general practitioner, the number of patients consulting for mental health issue increased during the first lockdown. This ratio rose to 72% during the second Lockdown.

Some of these elements were added to the discussion : “Furthermore, the reduction in consultations was less marked during the second lockdown, but the reduction in incidence of AD, bronchiolitis and ARI were almost the same than in the first lockdown. Only 8% of the general practitioner reported a 10 hours per week decrease in their volume of work after the first lockdown. During the second lockdown this ratio fall to 3%. According to 54% of the general practitioner, the number of patients consulting for mental health issue increased during the first lockdown. This ratio rose to 72% during the second lockdown [29], implying that consultations almost regained expected levels afterwards but disease incidence did not.”

---

## [Decision Letter · Decision Letter 1]

27 Sep 2021

Common communicable diseases in the general population in France during the COVID-19 pandemic

PONE-D-21-16521R1

Dear Dr. Launay,

We’re pleased to inform you that your manuscript has been judged scientifically suitable for publication and will be formally accepted for publication once it meets all outstanding technical requirements.

Kind regards,

Jeffrey Shaman

Academic Editor

PLOS ONE

Additional Editor Comments (optional):

Reviewers' comments:

Reviewer's Responses to Questions

**Comments to the Author**

1. If the authors have adequately addressed your comments raised in a previous round of review and you feel that this manuscript is now acceptable for publication, you may indicate that here to bypass the “Comments to the Author” section, enter your conflict of interest statement in the “Confidential to Editor” section, and submit your "Accept" recommendation.

Reviewer #1: All comments have been addressed

2. Is the manuscript technically sound, and do the data support the conclusions?

Reviewer #1: Yes

3. Has the statistical analysis been performed appropriately and rigorously? 

Reviewer #1: Yes

4. Have the authors made all data underlying the findings in their manuscript fully available?

Reviewer #1: Yes

5. Is the manuscript presented in an intelligible fashion and written in standard English?

Reviewer #1: Yes

6. Review Comments to the Author

Reviewer #1: The authors did a nice job responding to the reviewer suggestions. I have no further comments.

7. PLOS authors have the option to publish the peer review history of their article (what does this mean?). If published, this will include your full peer review and any attached files.

Reviewer #1: No

---

## [Editor Report · Acceptance letter]

1 Oct 2021

PONE-D-21-16521R1 

Common communicable diseases in the general population in France during the COVID-19 pandemic 

Dear Dr. Launay:

I'm pleased to inform you that your manuscript has been deemed suitable for publication in PLOS ONE. Congratulations! Your manuscript is now with our production department. 

Kind regards, 

on behalf of

Prof. Jeffrey Shaman 

Academic Editor

PLOS ONE